# Chronometric vs. Structural Hypercoagulability

**DOI:** 10.3390/medicina57010013

**Published:** 2020-12-28

**Authors:** Carmen Delianu, Mihaela Moscalu, Loredana Liliana Hurjui, Claudia Cristina Tărniceriu, Oana-Viola Bădulescu, Ludmila Lozneanu, Ion Hurjui, Ancuta Goriuc, Zinovia Surlari, Liliana Foia

**Affiliations:** 1Department of Biochemistry, “Grigore T. Popa” University of Medicine and Pharmacy, 700115 Iasi, Romania; carmendelianu@gmail.com (C.D.); ancuta.goriuc@yahoo.com (A.G.); georgeta.foia@umfiasi.ro (L.F.); 2Central Clinical Laboratory—Hematology Department, “Sf. Spiridon” County Clinical Emergency Hospital, 700111 Iasi, Romania; 3Department of Preventive Medicine and Interdisciplinarity, “Grigore T. Popa” University of Medicine and Pharmacy, 700115 Iasi, Romania; 4Department of Morpho-Functional Sciences II, Discipline of Physiology, “Grigore T. Popa” University of Medicine and Pharmacy, 700115 Iasi, Romania; violabadulescu@yahoo.com; 5Department of Morpho-Functional Sciences I, Discipline of Anatomy, “Grigore T. Popa” University of Medicine and Pharmacy, Universității str. 16, 700115 Iasi, Romania; claudia.tarniceriu@umfiasi.ro; 6Hematology Clinic, “Sf. Spiridon” County Clinical Emergency Hospital, 700111 Iasi, Romania; 7Department of Morpho-Functional Sciences I, Discipline of Histology, “Grigore T. Popa” University of Medicine and Pharmacy, Universității str. 16, 700115 Iasi, Romania; ludmila.lozneanu@umfiasi.ro; 8Department of Pathology, “Sf. Spiridon” Emergency County Hospital, 700111 Iasi, Romania; 9Department of Morpho-Functional Sciences II, Discipline of Biophysics, “Grigore T. Popa” University of Medicine and Pharmacy, 700115 Iasi, Romania; ion.hurjui@umfiasi.ro; 10Department of Odontology and Parodontology, “Grigore T. Popa” University of Medicine and Pharmacy, Universității str. 16, 700115 Iasi, Romania; zinovia.surlari@umfiasi.ro; 11Central Clinical Laboratory—Biochemistry Department, “Sf. Spiridon” County Clinical Emergency Hospital, 700111 Iasi, Romania

**Keywords:** stasis, endothelial damage, tissue factor, false hypercoagulability, chronometric

## Abstract

Prolonged tourniquet stasis induced by venepuncture can lead to the release of the plasma of cell lysis products, as well as tissue factor (TF), impairing the quality of coagulation test results. The accidental presence of TF in vitro can trigger the coagulation mechanism, generating a false decrease in prothrombin time (PT). *Background and Objectives:* Identification of short PT tests below the normal reference value that could suggest a situation of hypercoagulability. The study aimed to compare the results of the shortened PT tests at their first determination with the eventual correction following duplication of the analysis from the same sample. *Materials and methods:* Identification of the shortened PT tests has been carried out for a period of 4 months, upon 544 coagulation samples referred to the Hematology department of Sf. Spiridon County Clinical Emergency Hospital from Iasi, Romania. *Results:* Out of the 544 samples of which the results indicated a state of hypercoagulability, by repeating the determination from the same sample, for 200 (36.76%) PT tests (*p* = 0.001) the value was corrected, falling within the normal reference range. For 344 (63.24%) tests, the results suggested a situation of hypercoagulability. *Conclusions:* In order to guarantee the highest quality of the laboratory services, a proper interpretation and report of the patients’ results must be congruent and harmoniously associated to the actual clinical condition of the patient. Duplication of the PT determination from the same sample would exclude situations of false hypercoagulability and would provide significant improvement for the patient’s safety.

## 1. Introduction

Widespread hemostatic dysfunctions as well as therapeutic monitoring with anticoagulants are highlighted in vitro based on routine screening test, prothrombin time (PT), and partially activated thromboplastin time (aPTT) [1]. The quality of the collected biological product has a critical role in coagulation determinations. When error sources are not taken into account, monitoring the therapy can become discouraging and sometimes dangerous due to the lack of efficiency and the bleeding risk that can be reached out. In line with other studies, a percentage between 60 and 80% of the total errors come from the preanalytical phase, starting with the sampling of the biological product [2,3]. Lima-Oliveira and collaborators claim that the preanalytical phase “has been described as the dark part of the moon in the diagnostic process” [4]. One of the sources of error for hemostasis determinations is the use of the tourniquet, widely used to identify the proper vein for a safe venepuncture [5]. Prolonged stasis correlated with vascular endothelial damage for venepuncture accidentally promotes the release of TF (tissue factor) [1], historically acknowledged as “tissue thromboplastin”, having a procoagulant role both in vivo and as in vitro. This name has been assigned to define the plasma substance under the influence of which FII (prothrombin) turns into FIIa (thrombin) [6,7]. Subsequent to a vascular lesion, in vitro, the coagulation cascade is triggered in the presence of TF, considered thus as the main initiator of the mechanism by binding to FVII-pro-convertin (VIIa complex) and activating FIX (antihemophilic B) (FIXa being representative for the intrinsic pathway) and FX-Stuart (FXa being representative for the extrinsic pathway) [8].

According to current studies in the national specialized literature since 1981, Kondi specified the following: on the background of increased active endogenous thromboplastin activity “20% FII (prothrombin) is sufficient to maintain a state of hypercoagulability” [7,8,9]. Furthermore, Kondi specifies the factors that participate in the formation of prothrombin in the second phase and fibrin in the third phase, occur in about 15s and do not play a decisive role in the genesis of hypercoagulability, with Kondi defining it as “chronometric hypercoagulability”. The author may use this term based on the fact that the unit of measurement of PT is expressed in seconds. At that time, he mentioned that “prothrombin complex factors have a passive role, being activated by excess pro-thromboplastic substances”.

Informed consent was not necessary as the research was performed on samples from patients who had already been subjected to PT tests. For a better interpretation of the results, we repeated the PT determination in an attempt to exclude the accidental presence of TF in vitro.

The main objective regarded to identification of short PT tests below the normal reference value that could suggest a situation of hypercoagulability. The study aimed to compare the results of the shortened PT tests at their first determination with the eventual correction following duplication of the analysis from the same sample.

## 2. Materials and Methods

### 2.1. Description of the Method and Materials Used

The present study has been examined and approved by the Professional Ethics Committee of Sf. Spiridon County Clinical Emergency Hospital of Iasi (code: 4; date: 3 February 2017), Romania. This prospective cohort study has been carried out over approximately 4 months, between May and September, on a total of 544 coagulation samples, from which shortened PT test were obtained. All the samples referred to the Hematology Laboratory within the Sf. Spiridon County Clinical Emergency Hospital during the interval were collected and analyzed prospectively. Samples were taken by venepuncture in vacuum containers with sodium citrate, respecting the ratio of 1 part anticoagulant: 9 parts blood [10,11]. Plasma was separated subsequent to centrifugation of the whole blood, for 10 min at 4000 rotations/min. The determinations were performed on the ACL TOP 500 automatic analyzer, based on the increase in viscosity of the tested plasma, detected by photocolorimetry. Reagents from Instrumentation Laboratory Company, HemosIL RecombiPlasTin 2G were used. The criteria for inclusion in the study and the repetition of the test included samples from which the first determination indicated a PT shortening, correlated with a possible in vitro chronometric hypercoagulability. The second inclusion criterion to correlate the shortened PT results with a possible structural hypercoagulability in vivo, an analysis was made with the following parameters: leukocytes, erythrocytes, platelets, fibrinogen, D-Dimers, microbial flora, as well as direct and total bilirubin. As a rule, in our laboratory, the shortened results carefully require a post-analytical reverification to exclude the accidental presence of the clot in the sample sediment. The reverification procedure is performed by transvasation “passing” the blood from the primary sample into an anticoagulant free test tube, called “control” in which the blood is “passed”, in order to highlight a possible clot [12,13]. The exclusion criteria referred to the samples from which the results fell within the normal and therapeutic reference range. 

### 2.2. Statistical Analysis

All data were analyzed with SPSS version 24 (IBM Corporation, North Castle Drive, Armonk, NY, USA). Depending on the characteristics of the series of values, the statistical reference indicators were presented. Numerical variables were expressed as mean and standard deviation and qualitative variables were presented as absolute (*n*) and relative (%) frequencies. Significant differences in time-dependent parameters were determined based on the results of nonparametric tests (Mann–Whitney U test, Kruskal–Wallis test) due to their inhomogeneity, and in the case of category variables the Chi-square test (Pearson Chi-square test or Yates test) was applied. The parameter dependence was determined based on the correlation coefficient (Pearson correlation) corresponding to the linear regression. The accuracy assessment for the determination of the hypercoagulability based on the first PT determination was performed in accordance to the characteristic of the receiver operator characteristic (ROC) curve, Se and Sp, and ROC. The general accuracy of the determination method can be described by the area under the ROC (area under curve (AUC)); the larger the area, the better the evaluation criterion. The curve can be used to decide where the best compromise between the sensitivity and specificity of the method for determining hypercoagulability is based. The maximum probability of error (significance level) was considered *p*-value = 0.05 (5%), indicating with 95% confidence that the statistical decision is correct. 

## 3. Results

Before starting of the study, over approximately 4 months we identified all shortened tests of a total of 23,615 coagulation samples. The basis of this study was the repetition of the shortened PT test determination at the first determination in the same sample, in order to have a reliable interpretation of the results obtained. There were 544 tests with shortened results indicating a state of hypercoagulability. These were included in the study, meeting the criteria for inclusion in the group. The interpretation of the results was performed by framing the PT values in the normal reference interval (seconds):(a)normal → (10–14 s);(b)shortening/hypercoagulability (<10 s).

To determine the potential in vitro impact of TF on the inaccuracy of 544 PT tests corresponding to the 544 patients included in the study, two sets of determinations were carried out (Table 1):(i).first determination: reduced PT values;(ii).second determination: corrected PTs, which had the same shortened value after repeated determination in the primary sample, without recommending to repeat the sampling.

The mean PT was significantly higher (*p* < 0.001) after repeating the determination, increasing from 9.64 ± 0.3 (first determination) to 9.83 ± 0.4 (second determination). The variance of the values in the second determination was significantly higher than in the first determination (F_Levene_ = 69.08, *p* = 0.001), which was explained by the significant increase in PT values in the case of 200 samples. For the duplicated assessments, it was noticed the maintenance of a normal distribution of the values (Figure 1), but the variation range was translated from the range 8.3 to 9.9 towards 8.1 to 11.1.

The differences between the first and second determinations, described in (Figure 2), were also analyzed.

For the second determination, a change in values has been noticed for 452 samples (83.08%). For 102 specimens (18.75%) there was a decrease tendency, while for 350 samples (64.34%) an increment could be observed (Table 2, Figure 3).

The differences between the two determinations revealed an average value of 0.194 ± 0.31 for the samples in which there were decreases in the PT test and 0.339 ± 0.26 for the samples in which there were increases in the test (Figure 3). These increases led to obtaining 200 tests (36.76%) with normal values. 

To highlight potential factors that could influence the difference between the first and second determinations, we studied the result of univariate analysis of the correlation between differences in second prothrombin time and cellular elements, coagulation factor I (fibrinogen), and degradation products of fibrin (D-dimers) (Table 3). Also we studied the regression line describing the variance of the PT test difference according to leukocytes (WBC)*,* erythrocytes (RBC) and platelets (PLT) count (Figure 4A–C) and regression line describing the variance of the difference between the PT test according to the levels of fibrinogen and D-dimers (Figure 5A,B).

The correlative analysis pointed out a significant direct correlation between fibrinogen values and the difference in PT values at the second determination (r = 0.247, *p* < 0.001) and a significant inverse correlation between D-dimers and the difference recorded (r = −0.758, *p* < 0.001). These results explain the fact that for higher fibrinogen values, the value obtained at the second determination is significantly augmented, and for high D-dimer values, significant decreases in PT values are obtained at the second determination (Table 3).

The presence of microbial flora did not significantly affect the changes in prothrombin time (PT) (*p* = 0.987) (Table 4), the values being insignificantly lower in the second determination. However, it is observed that, in the case of the present flora, the variation of the PT test differences is significantly smaller compared to the variation of the PT test differences in the absence of the microbial flora.

The influence of the presence of clot on the PT determination was studied by comparative analysis according to the changes that occurred in the determination, as described in Table 5.

In the samples where the clot was present, fewer cases recorded PT values changes (38.9%), compared to the frequency of non-coagulating cases (84.6%) which exhibited significant changes in the second PT determination (*p* = 0.00002). Furthermore, the quantitative study of the changes that took place at the second determination was determined by the presence of the clot (Table 5).

The results acknowledged that the accidental presence of the clot in the sampled sediment did not significantly affect the change in PT. Thus, the frequency of cases that did not indicate changes in PT (61.11%) was significantly higher (*p* = 0.00001) compared to the incidence of cases with an increase in PT values (16.67%). The increase recorded for 16.67% of cases was low, maintaining its status of hypercoagulability in vitro (Table 5).

In the case of accidental presence of clot in erythrocyte sediment, an insignificant increase in PT was observed (mean: 0.13s ± 0.06), which maintained the state of hypercoagulability. Subsequent to sampling repeat, the increase in PT in the absence of a clot was significantly higher (*p* = 0.0257) compared to that recorded when the clot was present (Table 6). The accuracy of hypercoagulability was analyzed based on the first determination of PT, together with the sensitivity and specificity of the method. For this purpose, the results of the second determination were used as a “gold standard”. In this case, the positive predictive value was 63.24%, with a sensitivity of 72.97% and a specificity of 70%.

Analytical results may be impaired when coagulation tests are performed on jaundiced plasma, which results in increased bilirubin, thus impeding light transmission or interfering with optical absorbance (Table 7).

To evaluate the strength of PT discrimination on hypercoagulability, the receiver operator characteristic (ROC) curve was performed. The calculated AUC value was AUC = 0.774 (*p* < 0.0001, 95% CI: AUC → 0.7348–0.8138) (Figure 6).

## 4. Discussion

Laboratory medicine is considered to be at the heart of the patient safety solution, as it is involved in 60–70% of medical decisions, and 80–90% of all diagnoses are performed based on laboratory tests. Therefore, sources of error can have a major negative impact on patient care [14,15,16]. The permanent transparent and systematic supervision of the testing process encourages and promotes investigations when errors occur, helping to identify procedures and strategies for improving it [16,17]. Through this study we wanted to bring into attention the possible false in vitro shortening of the PT test that has no justification in the clinical context. 

Magnette and colleagues specify in their study that gargle stasis should not exceed 1 min [18]. Prolonged stasis can lead to activation of endothelial cells, increased fibrinogen, and other coagulation factors such as FVII, FVIII, and FIX [19,20,21]. It is accepted in the literature that, subsequent to vascular endothelium impairment, tissue thromboplastin is released; this binds to FVII by activating it FVIIa, triggering the coagulation mechanism [7]. For clot formation, only a 3% activated FII percentage is sufficient [9,22]. In his book, entitled “Clinical Laboratory”, Kondi mentioned that following the setting up of in vivo hypercoagulability, the leukocytes, erythrocytes, and platelets cellular elements can become thrombogenic factors: leukocytes by mass destruction, in cases of leucosis, under the influence of the administered therapies; erythrocytes in anoxia and marked hypotension, by the appearance of the “sludge” phenomenon, capillary circulation being difficult due to the sticking of erythrocytes between them, forming rolls; and inflated thrombocytosis (up to 800 × 10^3^/µL) which predisposes to thrombosis and hypercoagulability. 

Kondi also defines structural hypercoagulability that is not closely related to coagulation kinetics and depends on FXIII (fibrin stabilization factor), PLT, the structure and evolution of the clot in which fibrinogen participates as a substrate in terms of quantity and quality. “Chronometric hypercoagulability” occurs if there is an increase in endogenous thromboplastin activity. In this context, we could consider it as prolonged stasis-induced, as a result of the accidental in vitro passage, together with the endothelial damage, of the TF. 

In vitro activation of TF is performed in the presence of Ca2+ and factors such as accelerin, proconvertin, and Stuart factor (FX). FX activation is performed in the presence of Ca2+ in both coagulation paths, by the following activated factors:extrinsic: tissue thromboplastin (FIIIa) and proconvertin (FVIIa);intrinsic: anti-hemolytic A (FVIIIa) and anti-hemolytic B (FIX) [7].

In our research, we did not measure the plasma concentration of TF (procoagulant role). Based on previous studies, we assumed that the accidental presence in vitro could cause a false shortening of the PT test. The possible effects of in vitro TF on thrombin generation (FIIa) for the shortened PT test were assessed by repeating the measurement in the same sample after completion of the first determination. Furthermore, in order to establish the degree of correlation with the cellular elements, we considered the numerical parameters for leukocytes (WBC), erythrocytes (RBC), platelets (PLT) [6], fibrinogen (Fg), fibrin degradation products D-dimers, and the presence of microbial flora in the urine sediment analysis (routine examination indicated by clinicians on admission). Changes in hemostasis during toxic infectious shock states which predispose to the installation of hypercoagulability are known in the literature [6]. Microbial toxins such as those derived from *E. coli*, Piocianic, and Proteus act on platelets generating microthrombi formation [6]. Platelet aggregation is also influenced by Staphylococcus aureus exotoxins [6,23], performing a lytic action upon fibrinogen [6]. In our study, some patients presented urinary tract infection with Escherichia Coli. Considering that the accidental in vitro presence of TF could have a procoagulant role, determining the acceleration of coagulation in the presence of calcium ions, we repeated all the shortened tests, using the same sample. The mean PT was significantly higher (*p* < 0.001) after repeating the determination, increasing from 9.64 s ± 0.3SD (first determination) to 9.83 s ± 0.4SD (second determination) (Figure 2). The variance of the values in the case of the second determination was significantly higher than in the case of the first determination (F_Levene_ = 69.08, *p* < 0.001). There was a significant increase in PT values in 200 samples. Subsequent to analysis duplication, the maintenance of a normal distribution of values was noticed, but the variation amplitude was translated from the range PT: 8.3 s–9.9 s to PT: 8.1 s–11.1 s. No direct significant correlation was found between WBC, RBC, and PLT values and the PT test. Lippi and Guidi mentioned in their study that Fg and D-dimers are included in the routine test table, based on which anticoagulated therapy is controlled [24]. These tests are crucial in diagnosing haemorrhagic and thromboembolic conditions. 

The authors recall the recommendation that the accidental in vitro presence of TF may interfere with the determination by intrinsic activation [25]. However, in agreement with the report of Rosensos et al., they were able to demonstrate that standardization of sampling does not influence the reliability of the determinations [26]. The report mentioned the inclusion of Fg in the cardiovascular risk profile [27]. In this context, the determination of plasma Fg depends on possible sources of variability, even in the case of minimal differences of 5% [26]. Based on correlated evaluation, our study pointed out a significant direct correlation between fibrinogen levels and the difference of PT values at the second determination (r = 0.247, *p* < 0.001), and a significant inverse correlation between D-dimers and the difference recorded (r = −0.758, *p* < 0.001). Interestingly, for increased fibrinogen levels, the value obtained in the second determination was significantly higher. Based on these correlations, according to Kondi’s definition, we can speak of a structural hypercoagulability [6]. For high values of D-dimers, significant decreases in PT values were obtained at the duplicated assay. The regression line describes the variance of the PT test difference according to leukocytes (WBC), erythrocytes (RBC), and platelets (PLT) count. Starting with 2014, we implemented in our laboratory the post-analytical procedure of reverification by transvasation of the samples from which shortened results are obtained (PT < 10 s) and those reflecting bleeding risk for the patient (INR ≥ 5.5), in order to exclude the accidental presence of clot in the erythrocyte sediment [12,13]. Kondi mentioned that the factors FI, FII, FV, and FVIII are found only in plasma, and they are missing in serum. In vitro, the following plasma factors are involved in clot formation: fibrinogen, prothrombin, proaccelerin, and anti-haemolytic A [28]. The presence of serum in plasma determines, through the degree of consumption of these factors, the false prolongation of the tests [28,29,30]. In our case, for the samples in which the clot in the sediment has been noticed, the result was revoked, the non-conformity being explained as follows: “Sample with clot. Repeat sampling.” It was interesting to note that after the recommendation for resembling for PT assessment, the state of hypercoagulability was maintained. In the presence of clot, PT values changed significantly in fewer cases (38.89%) compared to the frequency of cases without clot (84.6%) in which changes in the second determination of PT (*p* = 0.00002) arose. This pattern was suggestive for a possible “chronometric hypercoagulability”, upheld even through a low percentage of prothrombin (20%) [6].

Our concern for acknowledging, identifying and supervising the sources of error in the pre-analytic phase began in 2015 [31]. We considered that the laboratory can contribute to their cutback through a management based on medical staff training [32]. The incidence of shortened PT tests (<10 s) has caught our attention in 2015, and this was the starting point for the documentation regarding the sources of error that could influence the determination [33]. By performing this research, we have revealed that there are significant correlations in vitro, suggesting that the PT test may be a parameter sensitive to TF. By repeating the determination from the same sample, the positive predictive value was 63.24% with a sensitivity of Se = 72.97% and a specificity of 70%, being considered certainly correct. To evaluate the discriminatory power of PT on hypercoagulability, according to the ROC curve, the calculated value of AUC was 0.774 (*p* < 0.0001, 95% CI: AUC → 0.7348–0.8138). Wan Azman and colleagues noted that a key challenge for clinical laboratories is to make a difference between in vivo and in vitro hemolysis, which is traditionally detected by post-centrifugal visual inspection of the specimen, while increased bilirubin levels are a typical sign of in vivo hemolysis. [34].

In our study, the quantitative analysis showed significantly higher values of TB (Total Bilirubin) (*p* < 0.001) and DB (Direct Bilirubin) (*p* = 0.0007) in cases where PT decreased significantly at the second determination, compared with the values recorded in situations where duplication either resulted in increased values or did not result in changes in PT levels.

Referring to the recent study by Wang and colleagues, we noted that in order to ensure medical safety, it is desired to expand knowledge by providing a system of self-check in terms of efficacy and specificity, based on clinical data, for coagulation tests [35]. In our research, self-verification was based on repeating the test determination from the same sample, evaluating also prior to centrifugation the quality of the specimen, related to the collected volume, hemolyzed plasma, and excluding by inversion both total and partial blood coagulation. It is interesting that Tripodi also mentions in his article that the coagulation of the test plasma can be activated by small amounts of phospholipids and tissue factor—considered procoagulant factor, considering that there are differences between the mechanism of in vitro coagulation vs. in vivo [36,37]. Randell and colleagues mention that it is a last chance to identify errors and verify the results obtained by the laboratory, before they become part of the electronic medical record of the patient [38]. 

We appreciate that the duplication of the test determination in the same sample provides safety and added value within the process efficiency and improvement of the results’ quality issued by our laboratory, allowing further focus on verification of abnormal results and emergency testing, this being in agreement with other studies [38,39]. Based on these findings, we intend to continue the study, to expand the database and analyze in detail the mechanisms of chronometric and structural hypercoagulability, which will cut down the estimation error as well as results’ accuracy. 

From the clinician’s point of view, a shortened value of PT leads to the escalation of investigations that may eventually prove unnecessary. It requires the repetition of the sample, which is sometimes a real discomfort for the patient and supports the clinical impact of our study, because 36.76% samples with shortened TP were corrected without the need to collect them again from patient. Without this correction the clinician could raise the suspicion of a structural hypercoagulability (in vivo) that would require expensive investigations to establish an etiology. The literature data [40,41,42] showed that a shortened PT can be correlated with an increased risk of thrombosis, which justifies escalating investigations to establish the etiology and clinical context in which this structural hypercoagulability occurs. Our study also showed a close correlation between D-dimers (thrombosis markers) and shortened PT at the second determination. For clinicians, it is essential to provide real results that relieve the patient of additional discomfort and avoid unnecessary investigations that cost money and time.

Finally, laboratory specialists are responsible for the reliability of the issued results, given that a false negative or positive result can lead to more expensive procedures and inevitably repeated examinations, not to mention the discomfort created to the patient [43,44]. The general principle of medical ethics is based on the good condition of the patient, and a management based on quality aids to increase his satisfaction. The accountability of medical staff is an important pawn in providing medical services that must meet the requirements, offer confidence and reduce the frequency of errors in the pre-analytical phase.

## 5. Conclusions

We expect these findings to contribute to the interpretation of the curtail results, particularly for the PT test. The accidental presence of TF with procoagulant role in vitro is decisive for obtaining normal values of the studied parameters. Our findings support duplication of the determination in the same sample, which could significantly influence the accuracy of the PT test. Our study tries to provide clinicians with key issues in the results’ interpretation, justifying the need for repetitive, confirmatory, and follow-up tests. It is also of interest to laboratory specialists who are responsible for accurate and precise results’ delivery, ensuring thus proper management of patients.

## Figures and Tables

**Figure 1 medicina-57-00013-f001:**
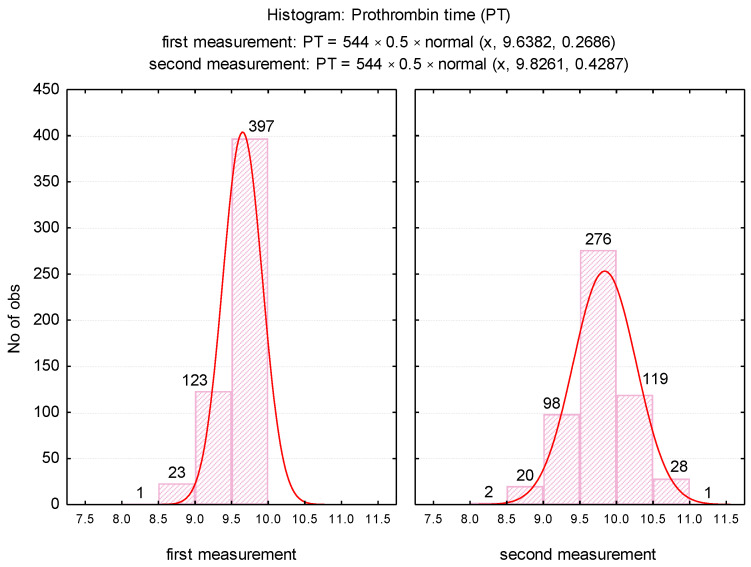
Histogram of PT values depending on the time of determination.

**Figure 2 medicina-57-00013-f002:**
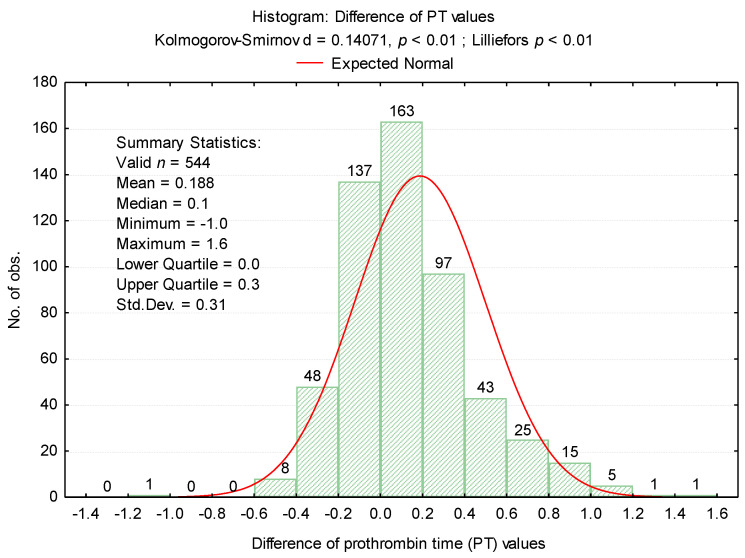
Histogram of differences between PT values.

**Figure 3 medicina-57-00013-f003:**
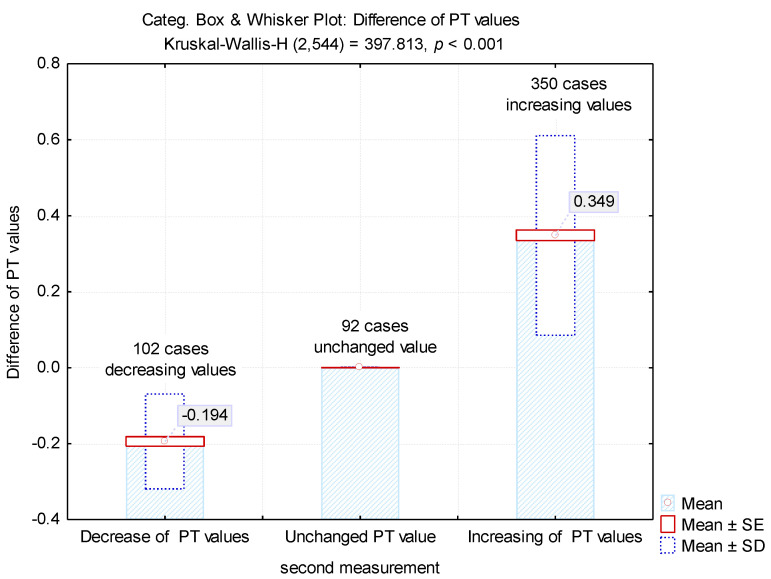
Mean values of the differences between the first and second determinations.

**Figure 4 medicina-57-00013-f004:**
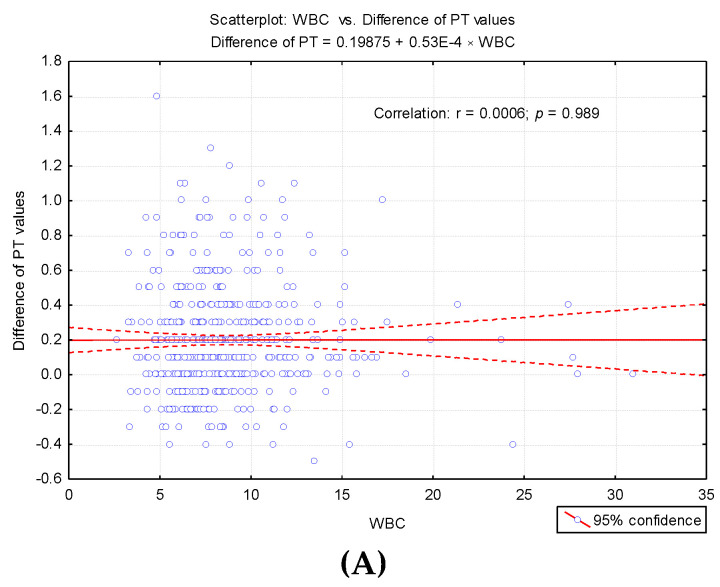
Regression line describing the variance of the PT test difference according to cellular elements count. (**A**) Regression line describing the variance of the PT test difference according to leukocytes (WBC) count. (**B**) Regression line describing the variance of the PT test difference according to erythrocytes (RBC) count. (**C**) Regression line describing the variance of the PT test difference according to platelets (PLT) count.

**Figure 5 medicina-57-00013-f005:**
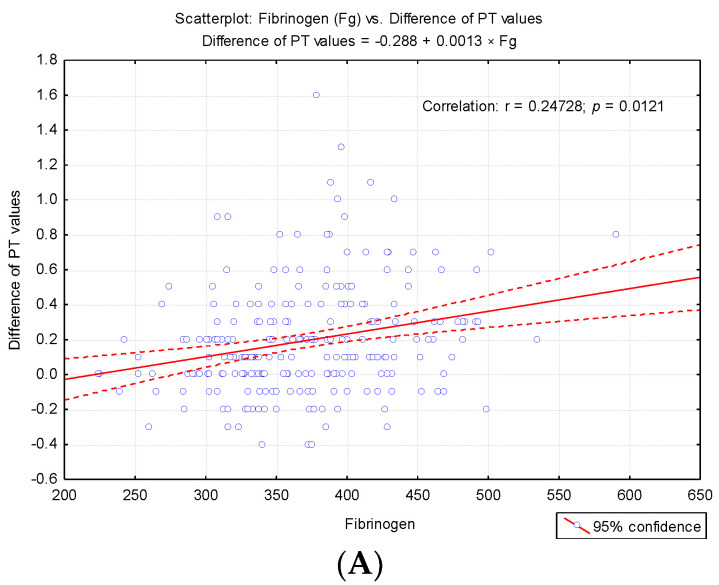
Regression line describing the variance of the difference between the PT test according to the level so fibrinogen (Fg) and D-dimers. (**A**) Regression line describing the variance of the difference between the PT test according to the levels of fibrinogen. (**B**) Regression line describing the variance of the difference between the PT test according to the levels of D-dimers.

**Figure 6 medicina-57-00013-f006:**
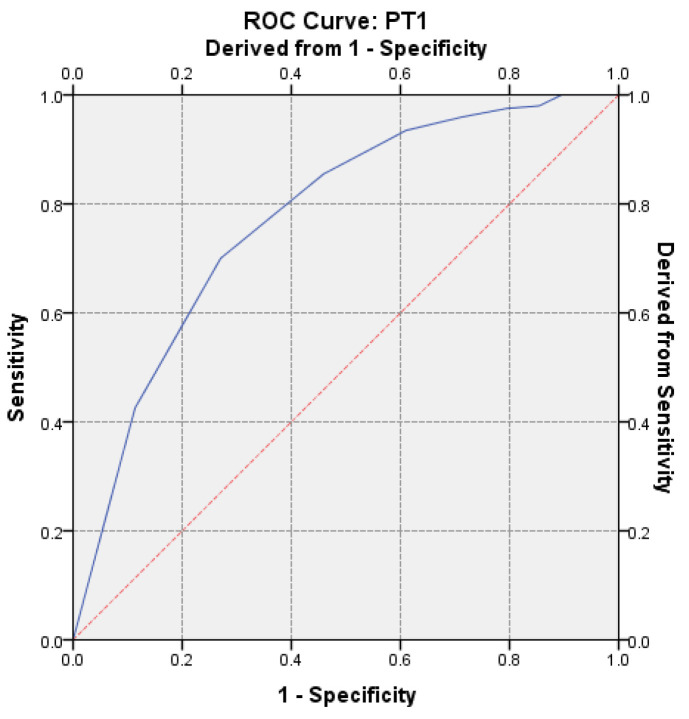
Receiver operator characteristic (ROC) curve on the assessment of hypercoagulability for the first determination.

**Table 1 medicina-57-00013-t001:** Prothrombin time (PT) values depending on the time of determination.

	The Moment of Determination		Statistic Test §	*p*-Value
	First Determination	Second Determination		
Prothrombin time (PT) (mean ± SD) median (min; max)	9.6 ± 0.3 9.7 (8.3; 9.9)	9.8 ± 0.4 9.8 (8.1; 11.1)	−8.3683	<0.001 *

Levene Test of Homogeneity of Variances: F = 69.08, *p* < 0.001 *; (§) Mann–Whitney U Test (*) Marked effects are significant at *p* < 0.05.

**Table 2 medicina-57-00013-t002:** Evaluation of hypercoagulability according to the second determination.

(*n* = 544)	Number of Tests	Percent
PT ≥ 10 s	200	36.76
PT < 10 (hypercoagulability)	344	63.23
Unchanged PT value	92	16.91
Modification of the value PT	452	83.08
Decrease of PT values	102	18.75
Increasing of PT values	350	64.33

PT—Prothrombin time.

**Table 3 medicina-57-00013-t003:** Correlation coefficients between the differences of the first and second determination and the cellular elements, fibrinogen, and D-dimers.

PT Difference (Second Determination—First Determination)	r (Correlation Coefficient) ‡	*p*-Value
Leukocytes (WBC)	0.0006	0.989
Erythrocytes (RBC)	0.0580	0.192
Platelets (PLT)	0.0362	0.416
Fibrinogen	0.2473	<0.001 *
D-dimers	−0.7580	<0.001 *

‡ Pearson correlation. (*) Marked effects are significant at *p* < 0.05.

**Table 4 medicina-57-00013-t004:** Mean values and statistical indicators of the difference between PT values at the second determination and microbial flora.

	Microbial Flora		Statistic Test §	*p*-Value
	Absence	Present		
Difference in PT values at the second determination (mean ± SD)	0.23 ± 0.39	0.19 ± 0.29	−0.016071	0.987178

Levene Test of Homogeneity of Variances: F = 4.83, *p* < 0.0297 *; (§) Mann–Whitney U Test (*) Marked effects are significant at *p* < 0.05.

**Table 5 medicina-57-00013-t005:** Accidental presence of clot in erythrocyte sediment interferes with determination of PT test. Quantitative evaluation of PT test changes in the presence of clot in erythrocyte sediment.

	Clot Absent (*n* = 526)	Clot Present (*n* = 18)	*p*-Value
**Changes in prothrombin time (PT)**			
no changes, *n*(%)	81 (15.4%)	11 (61.1%)	0.00002 *
present changes, *n*(%)	445 (84.6%)	7 (38.9%)	
**PT values in the second determination**			
Unchanged, *n*(%)	81 (15.4%)	11 (61.1%)	0.00001 *
Decrease, *n*(%)	98 (18.6%)	4 (22.2%)	
Growth, *n*(%)	347 (65.9%)	3 (16.7%)	

Yates Chi-square test, (*) Marked effects are significant at *p* < 0.05.

**Table 6 medicina-57-00013-t006:** Mean values of differences between the first and second determinations, depending on the presence of clots.

Difference in PT Values at the Second Determination	Clots		Statistic Test §	*p*-Value
	Absence	Present		
Decrease (mean ± SD)	−0.19 ± 0.10	−0.40 ± 0.41	9.3234	0.0257 *
Growth (mean ± SD)	0.35 ± 0.26	0.13 ± 0.06		

Levene Test of Homogeneity of Variances: F = 4.83, *p* < 0.0297 *; (§) Kruskal–Wallis H Test (*) Marked effects are significant at *p* < 0.05.

**Table 7 medicina-57-00013-t007:** Total Bilirubin (TB) and Direct Bilirubin (DB) values according to the evolution of the PT test at the second determination.

The Evolution of the PT Test at the Second Determination	Values of Total Bilirubin (TB)	Statistic Test §	*p*-Value
Decrease (mean ± SD)	3.94 ± 4.34	38.6731	0.00001 *
Unchanged (mean ± SD)	1.73 ± 0.25		
Growth (mean ± SD)	0.77 ± 1.09		
	**Values of Direct Bilirubin (DB)**		
Decrease (mean ± SD)	6.43 ± 4.97	14.6724	0.0007 *
Unchanged (mean ± SD)	0.23 ± 0.09		
Growth (mean ± SD)	0.31 ± 0.22		

Levene Test of Homogeneity of Variances: F = 4.83, *p* < 0.0297 *; (§) Kruskal–Wallis H Test (*) Marked effects are significant at *p* < 0.05.

## Data Availability

The data selected for the study are original and the aim is to improve the quality of possibly falsely short-ened PT tests.

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
