# Peer review of "Chronometric vs. Structural Hypercoagulability"

_medicina, 2020, doi:10.3390/medicina57010013_

Round 1

Reviewer 1 Report

The manuscript by Carmen Delianu et al. deals with the analysis of false positive shortened PT in common laboratory practice due to pre-analytical cause of hypercoagulability. Authors conclude that duplication of the PT determination would exclude situations of false hypercoagulability. The topic is of interest and deserves publication, the manuscript is well written. These are major points to be addressed:

1) Introduction, pg. 2 lines 58-62: the sentence should be moved to methods section, as it is not part of the background.

The aim of the study is not clear in this form, please clarify.  

2) Methods:  it is not clear the type of study. Is this a prospective cohort study? How 544 were selected? Were consecutive? I think that the number of overall samples considered should be described in the results section.  

Line 73-74 is part of the background-aim, not methods.

Criteria for “PT shortening” should be precisely clarified. Inclusion criteria were not mentioned.

Methods should include the list and the rationale for all analyses included (i.e. D-dimer, fibrinogen, cells count, bilirubin)

3) Results: Authors should clarified how many samples they considered overall and how many they included according to “PT shortening” criteria.

PT criteria must be moved to the methods section.

Data on PTT should also be included.

4) Discussion:  it is too long and should be focused on authors’ findings in the context of available literature. For example the paragraph on COVID-19 is unnecessary for the aim of this study and should be deleted.

Author Response

We made major changes to our paper, as the reviewers’ comments.

            The authors would like to thank the anonymous reviewers for their important observations and pertinent remarks, which lead us to substantially improve our paper.

Reviewer:,,1) Introduction, pg. 2 lines 58-62: the sentence should be moved to methods section, as it is not part of the background.

The aim of the study is not clear in this form, please clarify.”

Authors:

            In the sentence: ,,At that time, he mentioned that “prothrombin complex factors    have     a passive role, being activated by excess pro-thromboplastic substances.”   Through this phrase, the author (Kondi) explains which are the substances with the  activating role of the prothrombin complex, as a continuation of the previous sentence.

            ,,The present study has been examined and approved by the Professional Ethics Committee of Sf. Spiridon County Clinical Emergency Hospital of Iasi(code: 4; date: 3 February 2017), Romania.” has been moved to the methods section.

            The purpose of the study was corrected by the established objectives.

Reviewer:,,2) Methods:  it is not clear the type of study. Is this a prospective cohort study? How 544 were selected? Were consecutive? I think that the number of overall samples considered should be described in the results section. 

Line 73-74 is part of the background-aim, not methods.

Criteria for “PT shortening” should be precisely clarified. Inclusion criteria were not mentioned.

Methods should include the list and the rationale for all analyses included (i.e. D-dimer, fibrinogen, cells count, bilirubin).”

Authors:

            The type of study was defined: prospective cohort, as well as the role of the 544    samples analyzed. During the 4 months, 544 shortened PT tests were identified which were compared with the results obtained from the same sample by repeating the  determination. In the results section, we described the criterion for including the PT tests analyzed according to the recommendation.

We consider line 73-74 to be the description of the material used to determine the coagulation tests.

            We defined the inclusion criteria for in vitro chronometric and structural     hypercoagulability in vivo according to the recommendations in the "Materials and  methods" section, also justifying the choice of correlation with D-Dimers.

            These parameters were chosen to demonstrate if there is a correlation between a shortened PT knowing that inflammation, infection, polyglobulia are associated with structural hypercoagulability and D-dimers are markers of thrombosis.

Reviewer:,,3) Results: Authors should clarified how many samples they considered overall and how many they included according to “PT shortening” criteria.

PT criteria must be moved to the methods section.

Data on PTT should also be included.”

Authors:

According to the recommendations, we entered in the "Results" section the total number of samples sent to our laboratory to determine the coagulation tests during the 4-month period in which the study was conducted (a total of 23615 coagulation samples).The criteria for including the analyzed PT tests and for excluding the PT tests are mentioned in the section "Material and methods".

The study aimed to update the term chronometric hypercoagulability in vitro and correlate shortened PT tests with structural hypercoagulability in vivo; the analysis and correlation with PTT is up to the clinician.

Reviewer:,,4) Discussion:  it is too long and should be focused on authors’ findings in the context of available literature. For example the paragraph  on COVID-19 is unnecessary for the aim of this study and should be deleted.”

Authors:

            According to the recommendation, the paragraph on COVID-19 has been deleted. We introduced discussions on the clinical impact of the study results with reference to data from the literature (reference 40 - 42). Consequently reference 49 and 50 became reference 43 and 44.

Reviewer 2 Report

Lines 97, 98: ‘(AUC - Area under Curve)’ defined twice?

Lines 112/113: “second determination: corrected PTs, which had the same shortened value after repeated determination in the primary sample, without recommending to repeat the sampling.” Has unclear meaning

Table 1: The difference between PTs of 9.83 and 9.64 may be statistically significant, but does not have any clinical significance that I can determine, especially as clinical results are generally reported to whole numbers.

If the sample was just retested, how can this then point to TF release?

Figure 2: it is unsurprising that the majority of samples increased slightly, since the PTs were short to begin with; it is harder for short PTs to become shorter!

Table 2: ditto: as the samples mostly had PTs just under 10s, it is not surprising to see many of them increase to >10s.

Fig 6: what data was used to calculate the ROC AUC?

Author Response

We made major changes to our paper, as the reviewers’ comments.

            The authors would like to thank the anonymous reviewers for their important observations and pertinent remarks, which lead us to substantially improve our paper.

Response to reviewers       

Reviewer:,,Lines 97, 98: ‘(AUC - Area under Curve)’ defined twice?”

Authors:

We corrected in the manuscript the definition of the area under the curve twice.

Reviewer:,,Lines 112/113: “second determination: corrected PTs, which had the same shortened value after repeated determination in the primary sample, without recommending to repeat the sampling.” Has unclear meaning.”

Authors:

We wanted to mention the aspect that the retest of short PT-test were performed from the same sample without repeat sampling.

Reviewer:,, Table 1: The difference between PTs of 9.83 and 9.64 may be statistically significant, but does not have any clinical significance that I can determine, especially as clinical results are generally reported to whole numbers.”

Authors:

Yes, even if apparently between 9.64 and 9.83 is a small difference (observational) it is statistically significant (p <0.05). The sample size is large which increases the statistical power of the estimates. Because we have a numerical series of values without normal distribution we applied for comparison the Mann-Whitney U Test, specific for these situations. The results are correct.

This reflects that a part of samples increased the value of PT, being false short values. These false results of PT no tallow to clinicians a right diagnosis of a structural hypercoagulability that leads to the escalation of investigations that may eventually prove unnecessary. It requires the repetition of the sample, which is sometimes a real discomfort for the patient.

Reviewer:,,If the sample was just retested, how can this then point to TF release?”

Authors:

Based on the literature, a possible acceleration of the in vitro coagulation mechanism could be the accidental presence of tissue factor. That is why we mentioned in this study that the concentration of the tissue factor was not determined.

Reviewer:,,Figure 2: it is unsurprising that the majority of samples increased slightly, since the PTs were short to begin with; it is harder for short PTs to become shorter!”

Authors:

It is one of the things that made us think. We propose a future study on a larger sample in which to analyze this aspect as well.

Reviewer:,,Table 2: ditto: as the samples mostly had PTs just under 10s, it is not surprising to see many of them increase to >10s.”

Authors:

In this context, based on the literature, we considered that this correction represents precisely the chronometric hypercoagulability based on the kinetics of the in vitro coagulation mechanism.

Reviewer:,,Fig 6: what data was used to calculate the ROC AUC?”

Authors:

ROC was performed to evaluate the predictive power of PT on hypercoagulability.

Round 2

Reviewer 1 Report

Authors addressed all the issues. 

Reviewer 2 Report

no more suggestions